# The protein interactome of *Escherichia coli* carbohydrate metabolism

**Shomeek Chowdhury**[1], **Stephen S. Fong**[1], **Peter Uetz**[2]*

1 Center for Integrative Life Sciences Education, Virginia Commonwealth University, Richmond, VA, United States of America, 2 Center for Biological Data Science, School of Life Sciences, Virginia Commonwealth University, Richmond, VA, United States of America

* uetz@vcu.edu

**Data Availability Statement:** All relevant data for this study are within the paper and its Supporting Information files.

**Funding:** The author(s) received no specific funding for this work.

## Abstract

We investigate how protein-protein interactions (PPIs) can regulate carbohydrate metabolism in *Escherichia coli*. We specifically investigated the stoichiometry of 378 PPIs involving carbohydrate metabolic enzymes. In 48 interactions, the interactors were much more abundant than the enzyme and are thus likely to affect enzyme activity and carbohydrate metabolism. Many of these PPIs are conserved across thousands of bacteria including pathogens and microbial species. *E. coli* adapts to different cellular environments by adjusting the quantities of the interacting proteins (25 PPIs) in a way that the protein-enzyme interaction (PEI) is a likely mechanism to regulate its metabolism in specific environments. We predict 3 PPIs (RpsB-AdhE, DcyD-NanE and MinE-Yccx) previously not known to regulate metabolism.

## 1 Introduction

Carbohydrate metabolism (CHM) is responsible for converting sugars and other compounds into other metabolites and ATP and occurs in all organisms from bacteria to humans [1]. Carbohydrate metabolism includes about a dozen sub-pathways in bacteria and thus can involve more than a hundred metabolic enzymes [1]. In *E. coli*, the focus of this study, CHM involves 13 sub-pathways (**Table 1** and **Fig 6**).

Central carbon metabolism is the shortest route for producing energy and biomass in *Escherichia coli* [3]. In addition, CHM provides the substrates for many other metabolic pathways, such as lipid or amino acid metabolism [2, 4]. Given that carbohydrate and energy metabolism in general is critical for cellular survival, it is usually tightly regulated [3]. Different regulatory mechanisms have been identified, including gene expression [5], post-translational modification (PTMs) [6], enzyme localization [7], or allosteric control [8]. In addition, protein-protein interactions (PPIs) are a common mechanism to regulate protein activity. Although numerous enzymes have been shown to be regulated by PPIs, it is difficult to study the impact of PPIs on metabolism on a large scale. For instance, we have demonstrated how several metabolic enzymes in *Escherichia coli* are regulated by PPIs, including enzymes in energy metabolism, carbohydrate metabolism or amino acid metabolism [8–11]. Notably, a single protein such as Hpr can both activate and repress enzymes in the same pathway such as glycolysis [10]. It is well-known that many enzymes are regulated by interactions in large regulatory networks [12].

**Competing interests:** The authors have declared that no competing interests exist.

**Table 1. Number of enzymes of *E. coli* carbohydrate metabolism pathways based on KEGG [2].**

| KEGG Identifier | Metabolic Pathway | Enzyme Count |
|---|---|---|
| 10 | Glycolysis/Gluconeogenesis | 26 |
| 20 | Citrate Cycle (TCA cycle) | 19 |
| 30 | Pentose phosphate pathway | 18 |
| 40 | Pentose and glucoronate interconversions | 6 |
| 51 | Fructose and mannose metabolism | 13 |
| 52 | Galactose metabolism | 14 |
| 53 | Ascorbate and aldarate metabolism | 5 |
| 500 | Starch and sucrose metabolism | 13 |
| 520 | Amino sugar and nucleotide sugar metabolism | 20 |
| 620 | Pyruvate metabolism | 31 |
| 630 | Glyoxylate metabolism and dicarboxylate metabolism | 24 |
| 640 | Propanoate metabolism | 18 |
| 650 | Butanoate metabolism | 23 |
| 660 | C5-Branched dibasic acid metabolism | 10 |
| **562** | Inositol phosphate metabolism | 2 |
| **Total** | | **242** |

Building upon a previous pilot study on glycolysis [13], here we investigate the connection between interactomics (PPI networks) and metabolomics (metabolic networks) on a more comprehensive scale, focusing on carbohydrate metabolism (CHM). We wondered to what extent carbohydrate metabolism can be regulated via protein-enzyme interactions of its enzymes [14] and how many such interactions are there for different carbohydrate pathways [15]. For example, if a protein binds an enzyme metabolizing inositol phosphate, that protein-enzyme interaction (PEI) will be considered as part of the inositol phosphate pathway. Central carbohydrate pathways such as glycolysis, TCA cycle, pyruvate metabolism and HMP shunt have a high number of PEIs which is not surprising as these are primary fueling pathways with high number of enzymes [16–18]. Importantly, a good number of uncharacterized proteins also bind carbohydrate enzymes, possibly shedding light on novel metabolic regulatory mechanisms [19, 20].

Proteins interacting with carbohydrate metabolic enzymes may perform important functions as these *E. coli* PEIs can be predicted in more than 4000 bacterial genomes (40% of the known genomes). Such potentially conserved interactions are good candidates for future experimental analysis and thus may reveal novel regulators of metabolism [21, 22] (**Table 3**). Therefore, a further understanding of these carbohydrate metabolism PEIs may contribute to applications such as biofuel production [23–25], antibiotic development [11, 26] or comprehending metabolism functions of the human microbiome [27].

In this study, we identified 25 PEIs which likely perform metabolic regulatory roles based on the abundance of enzymes and their interacting proteins [28]. We verify this by using published protein abundance data of 2600 *E. coli* proteins across 22 environmental conditions, including multiple carbon sources [29].

## 2 Materials and methods

### 2.1 Metabolic enzymes and their pathway information

We obtained the list of metabolic genes of carbohydrate metabolism divided into different carbohydrate pathways from KEGG [2, 30] (**Table 1**, for full list see **S1 Table in S1 File**). These

genes were mapped onto proteins of *E. coli* K-12 (proteome ID: UP000000625) using Uni-Prot's ID mapping feature [31]. This resulted in a list of all *E. coli* metabolic enzymes categorized into 15 carbohydrate metabolic pathways.

## 2.2 Protein-enzyme interaction data

We used the IntAct database to obtain *E. coli* protein-protein interactions (PPI) [32]. The PPIs in IntAct are annotated by interaction type, detection method and experimental evidence giving them a PPI score. If the PPI score exceeds a certain threshold based on these parameters, it is included in the database with the literature evidence information [33, 34].

IntAct divides the PPIs into spoke and non-spoke expanded interactions. Spoke-expanded interactions are PPIs obtained from protein complexes from which binary PPIs are inferred without knowing whether these interactions physically take place [32, 35]. Hence, we focused on non-spoke expanded interactions that is, primarily binary protein-protein interactions whether they are from (binary) protein complexes or other sources. From this set, we chose the PPIs with at least one of the two proteins being a metabolic enzyme. A list of *E. coli* metabolic enzymes was already obtained earlier from KEGG and UniProt (**Section 2.1**). This list of protein-enzyme interactions (PEIs) had a total of 299 binding partners ("interactors") and 378 interactions (**S1 Table in S1 File**).

## 2.3 Inferring what kind of proteins bind metabolic enzymes

Interactors were classified into enzymes, metabolic proteins, and other (un-) characterized proteins. UniProt provided the EC number for proteins which are enzymes [36]. We used this feature to classify the interactors into enzymes and non-enzymes first. Then ToppGene (a gene enrichment software [37]) was used to classify the enzyme and non-enzyme interactors into metabolic and non-metabolic (**Fig 1**). The list of interactors was then processed in Topp-Gene to produce a list of Gene Ontology (GO) Biological Process terms along with the interactors falling under that GO term [38]. Only the "Metabolic Process" GO terms were chosen. If the interactor listed in these GO terms was an enzyme, it was called 1) metabolic enzyme (Acetyl-CoA carboxylase, Enolase); the remaining enzymes were called 2) non-metabolic enzymes (e.g., Multiphosphoryl transfer protein, ATP-binding protein) (**Figs 1 and 5**). In the same way, we classified the non-enzyme interactors into 3) metabolic (e.g., Cytochrome YdhU, Cellulose biosynthesis protein) and 4) non-metabolic proteins (e.g., Biopolymer transport protein, Inner membrane protein). Interactors whose protein names mention "Uncharacterized" (e.g., Fimbrial-like protein) or lacked GO annotations (e.g., Electron transport protein) were collectively called "unknown proteins".

## 2.4 Visualizing carbohydrate metabolism using KEGG mapper

In order to put carbohydrate metabolism in a broader context, we highlighted CHM in *E. coli*'s metabolome (KEGG: map01100) in **Fig 9**. The figure was constructed using KEGG mapper and each of *E. coli*'s 15 different carbohydrate metabolic pathways were color-coded separately [2, 39].

## 2.5 Conservation of *E. coli* PEIs

UniProt and EggNog provided clusters of ortholog (COG) group IDs. This COG group lists all the bacterial ortholog (OG) species where that protein is found as an ortholog [40]. We obtained that list of OG species from EggNOG for all *E. coli* enzymes and their interactors. Then, for each PEI, we counted the number of OG species in which both the enzyme and its

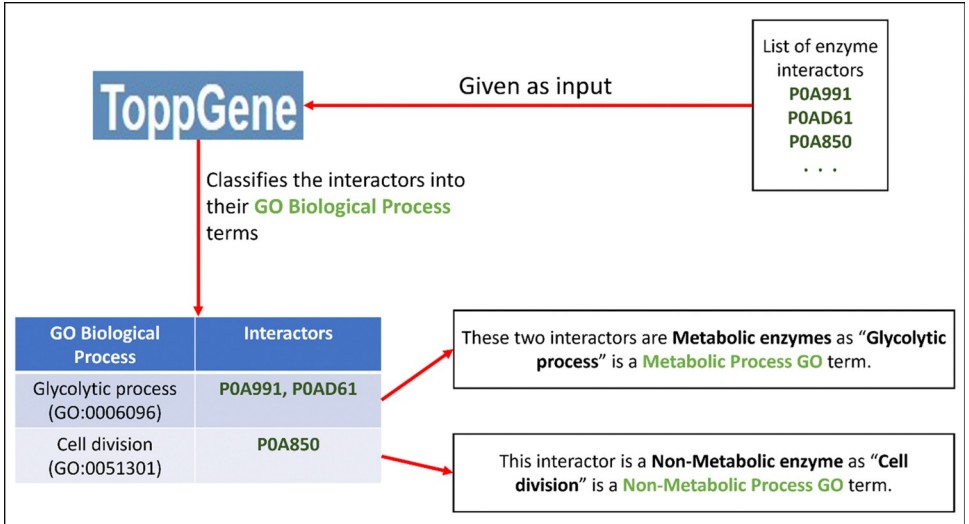

**Fig 1. Classification of enzyme interactors into metabolic and non-metabolic.**

interactor co-occurs and we called this number "conservation value" (**Figs 2 and 8**). **S1 Table in S1 File** lists the conservation values for all the *E. coli* PEIs. Presence of the orthologs of both the enzyme and interactor from *E. coli* is likely to indicate a conserved interaction in that ortholog species [41]. We have also listed some bacterial species known to cause diseases in **Table 3** where many *E. coli* PEIs (or rather predicted PEIs) are conserved.

## 2.6 Protein abundance of *E. coli* protein-enzyme interactions

Schmidt et al. experimentally determined the abundance of ~2600 *E. coli* proteins in 22 different growth conditions [29], measured as protein copies per cell (**Figs 3, 9, 11, 14 and S1 Table in S1 File**). We used these values to determine the *in vivo* ratios of PEIs.

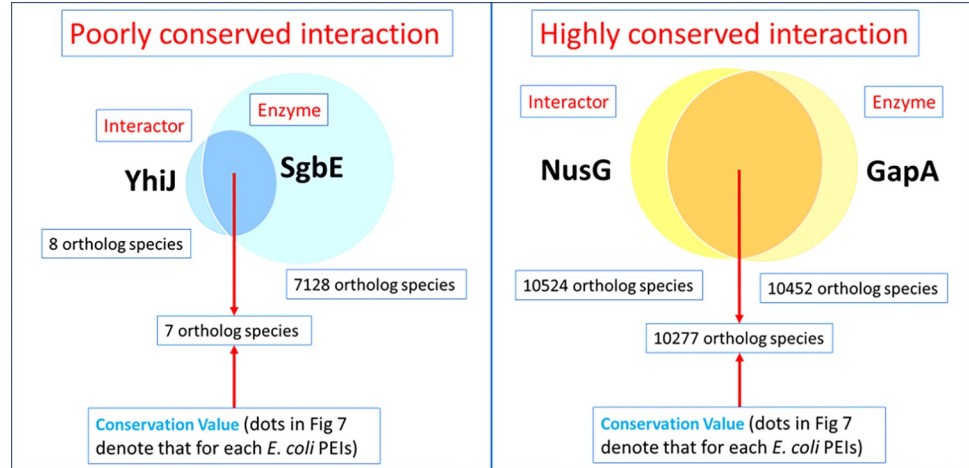

**Fig 2. The conservation value as a measure of PEI conservation.** YhiJ-SgbE is an example of a poorly conserved interaction with YhiJ present in 8 genomes and SgbE present in 7128 genomes, hence both proteins are found in only 7 genomes (the conservation value, CV, thus being 7). Similarly, the CV of NusG-GapA is 10,277, given that orthologs of both proteins are found in 10,277 genomes. Note that these values represent *predicted PPIs*.

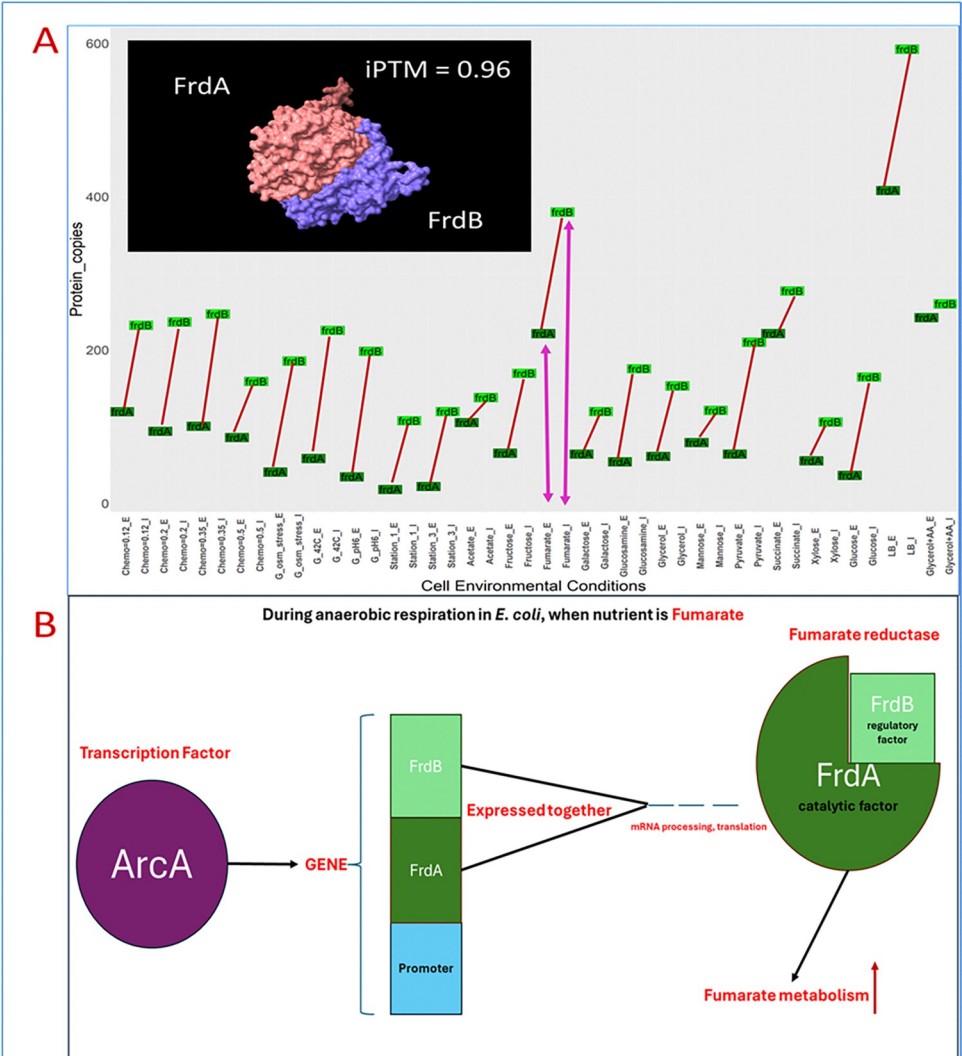

**Fig 3. FrdA and FrdB interaction.** In specific growth media (A), *E. coli* transcription factor ArcA will increase the expression of co-regulated genes FrdB and FrdA **(B)** and thus the enzyme complex (fumarate reductase) which metabolizes fumarate **(A)**. The interactor FrdB becomes the regulatory factor which allosterically regulates the activity of the metabolic enzyme FrdA. In x-axis, G_ means glucose media, so, G_osm_stress is glucose media with high NaCl concentration. _E and _I for each media in x axis are the same media but _E shows abundance of enzyme and _I shows abundance of interactor in that media.

## 2.7 Obtaining protein structures, identifying contact residues at protein-protein interfaces, and calculating their fractions

Single protein structures as well as protein-protein interaction complexes in **Section 3.7** and **3.8** were obtained with the help of ColabFold [42]. This is an AlphaFold-based protein structure software. The Predicted Template Modeling (PTM) and (Interface PTM) iPTM scores [43] mentioned for each modeled protein and protein-enzyme structure respectively, range from 0 to 1, indicating high quality and reliability of the predicted 3D structure if a score is > = 0.95.

ChimeraX software was then used to visualize the predicted protein structures, and identify the binding site residues at PPI interfaces [44]. This software considers two atoms in contact if they are within a 4 Å distance [45]. Therefore, with the help of ChimeraX, we determined

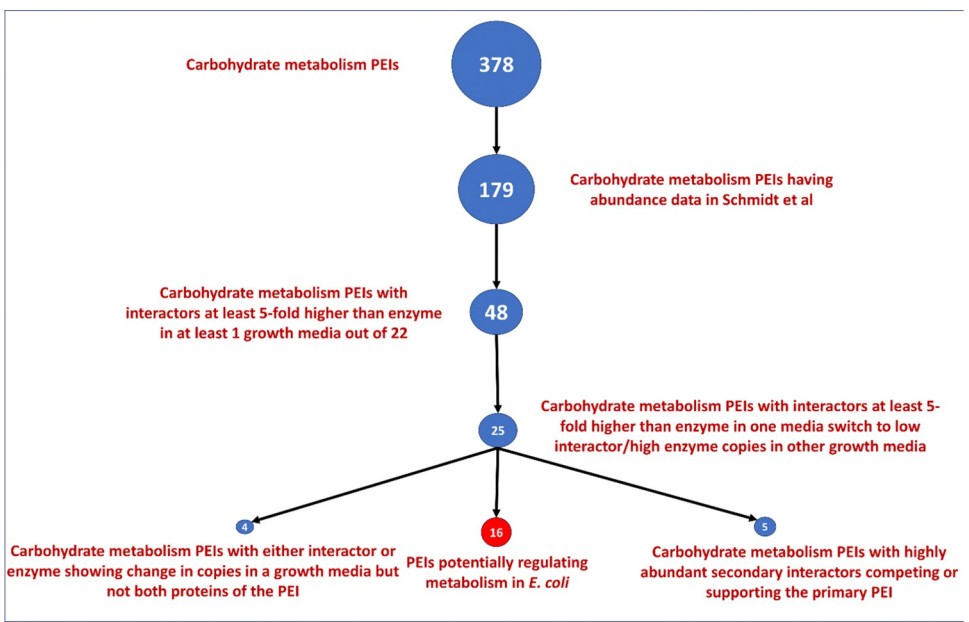

**Fig 4. Identification of 16 PEIs as metabolism regulators among 378 PEIs of *E. coli* carbohydrate metabolism.**

which residues are in contact (interface residues) between two proteins [48]. Once we obtained those, we calculated the fraction (%) of overlapping binding sites between secondary interactors, primary interactors and metabolic enzymes (**Section 3.7**) using the structures from ColabFold [46, 88]. We first counted how many residues are involved in the primary interaction, that is, interface residues between the primary interactor and the enzyme. After that, we

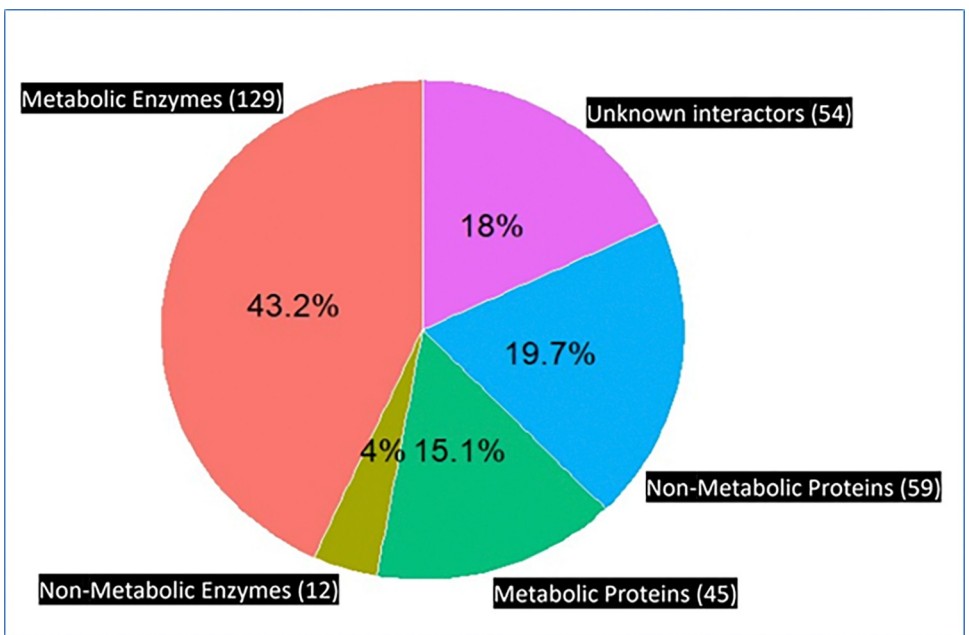

**Fig 5. *E. coli* carbohydrate enzymes primarily interact with other metabolic enzymes.** Shown are the 299 interactors of carbohydrate metabolism. They also interact with 104 non-enzymes comprising 45 metabolic and 59 non-metabolic proteins. *E. coli* carbohydrate enzymes also bind 54 proteins of unknown function.

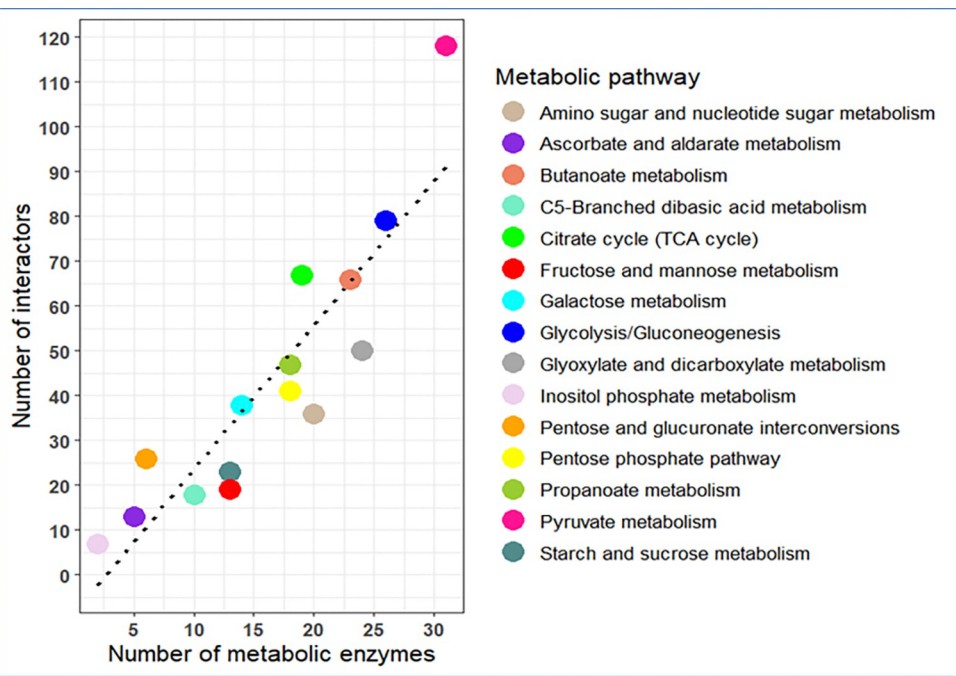

**Fig 6. Interactions in *E. coli* carbohydrate pathways correlate with pathway size.** Pathways with more enzymes bind more proteins compared to pathways with low count of enzymes.

**Table 2. Statistics for metabolic pathways, enzymes, and their interactions in *Escherichia coli*. For a complete list of proteins see S1 Table in S1 File.**

| Component | Count |
|---|---|
| Carbohydrate metabolic pathways | 15 |
| Carbohydrate metabolic enzymes | 242 |
| Carbohydrate enzymes binding interactors | 146 |
| Interactors binding carbohydrate enzymes | 299 |
| Carbohydrate metabolism protein-enzyme interactions (PEIs) | 378 |

**Table 3. Selected bacterial genomes with large numbers of orthologous PEIs.**

| Category | Species/ genome | Count of *E. coli* PEIs |
|---|---|---|
| *Pathogens* | *Escherichia coli* O157:H7 str. EDL933 | 355 |
| | *Shigella sonnei* 53G | 354 |
| | *Citrobacter koseri* ATCC BAA-895 | 337 |
| | *Salmonella enterica* subsp. *diarizonae* | 337 |
| | *Klebsiella oxytoca* | 331 |
| *Human microbiome* | *Pseudomonas aeruginosa* PAO1 | 256 |
| | *Haemophilus influenzae* Rd KW20 | 186 |
| | *Staphylococcus aureus* | 174 |
| | *Staphylococcus epidermidis* RP62A | 163 |
| | *Enterococcus faecalis* V583 | 153 |
| *Industrial models* | *Pseudomonas putida* W619 | 249 |
| | *Streptomyces roseochromogenus* | 221 |
| | *Bacillus subtilis* | 199 |
| | *Clostridium saccharoperbutylacetonicum* | 193 |
| | *Lactobacillus composti* | 175 |

**Table 4.  *E. coli* PEIs possibly regulated by secondary interactors based on abundance.**  Int1, Int2 = primary interactor, secondary interactor of enzyme. G: L = abundance in glucose and LB media (a single value indicates that only abundances for media with glucose are available, e.g. in the case of GlgX).

| Case | Enzyme | G:L | Interactor | G:L | Int2 | G:L | Int2 of Int1 | G:L |
|------|--------|-----|-----------|-----|------|-----|-------------|-----|
| 1 | GlgX | 111 | YiaD | 344 | MsrA | 687 | EpmB | 53:89 |
| 1+2 | IlvI | 441:114 | RimM | 776:1786 | IlvH | 1533 | ProQ | 2318: 4624 |
|  |  |  |  |  | MqsA | 34 | IlvF | 220:67 |
|  |  |  |  |  | LeuB | 4588 | RpsS | 19008:51051 |
|  |  |  |  |  |  |  | UbiH | 63:114 |
| 2 | FdoH | 4 | FdoG | 13 | N/A |  | FdhE | 374 |
| 2 | FucO | 41:234 | Def | 886:1542 | N/A |  | Usg | 456: 583 |
|  |  |  |  |  |  |  | UbiF | 31:46 |
|  |  |  |  |  |  |  | TrmH | 25:70 |
|  |  |  |  |  |  |  | RplD | 21084: 49874 |
|  |  |  |  |  |  |  | RplQ | 18398: 46700 |
|  |  |  |  |  |  |  | RplV | 49717: 137404 |
| 2 | GloC | 842 | DcrB | 2176 | N/A |  | OsmY | 10490 |

determined how many of these residues are also involved in the secondary interaction, that is, interface residues between secondary interactor and primary interactor (or enzyme). We report these fractions (%) of overlapping binding residues between primary and secondary interactions in **Table 5** and **Fig 12**.

## 2.8 Calculating buried surface area and binding energy

The buried surface area (BSA) is the surface area of proteins which is found at the PPI interface [46]. Before calculating BSA, first, we obtained structures of individual proteins making PPIs from ColabFold [42]. Then, by loading the structures (PDB files) of the binding proteins and then of the PPI complex in PyMol, we obtained the surface area of those proteins (or PPIs) structures by using the "get_area" PyMol command [47]. Once we have the surface area (SA) of 1) two interacting proteins individually and 2) PPI complex, BSA is the sum of surface areas of two proteins excluding surface area of the protein-protein complex. For example, the BSA of secondary interaction RplD-Def = SA of RplD + SA of Def–SA of RplD-Def complex. We calculated the buried surface area (BSA) between secondary interactor and primary interactor/ enzyme (secondary interaction) and between primary interactor and metabolic enzyme (primary interaction). Then, we compared those values to find which interaction (primary or secondary) has the bigger BSA (**Fig 12**) [47].

We finally determined the binding energy (ΔG) between two proteins [48] using the PRODIGY server [49], were inputting the two PDB files for two proteins binding each other outputs the ΔG value of their interaction. We used the server to determine the ΔG value of

**Table 5.  Secondary proteins-affected PEIs based on overlapping binding sites between the proteins involved in the interactions.**

| Protein-Enzyme Interaction (PEI) | Secondary Interactor Binding Enzyme (Case 1) | Secondary Interactor Binding Primary Interactor (Case 2) | Fraction of overlapping contact residues | Media |
|---|---|---|---|---|
| YiaD-GlgX | MsrA | N/A | 57.1% | Glucose |
| RimM-IlvI | LeuB | RpsS | 61.3%/100% | Glucose/ LB |
| FdoG-FdoH | N/A | FdhE | 16.3% | Glucose |
| Def-FucO | N/A | RplD | 23.8% | Glucose/ LB |
| DcrB-GloC | N/A | OsmY | 42.9% | Glucose |

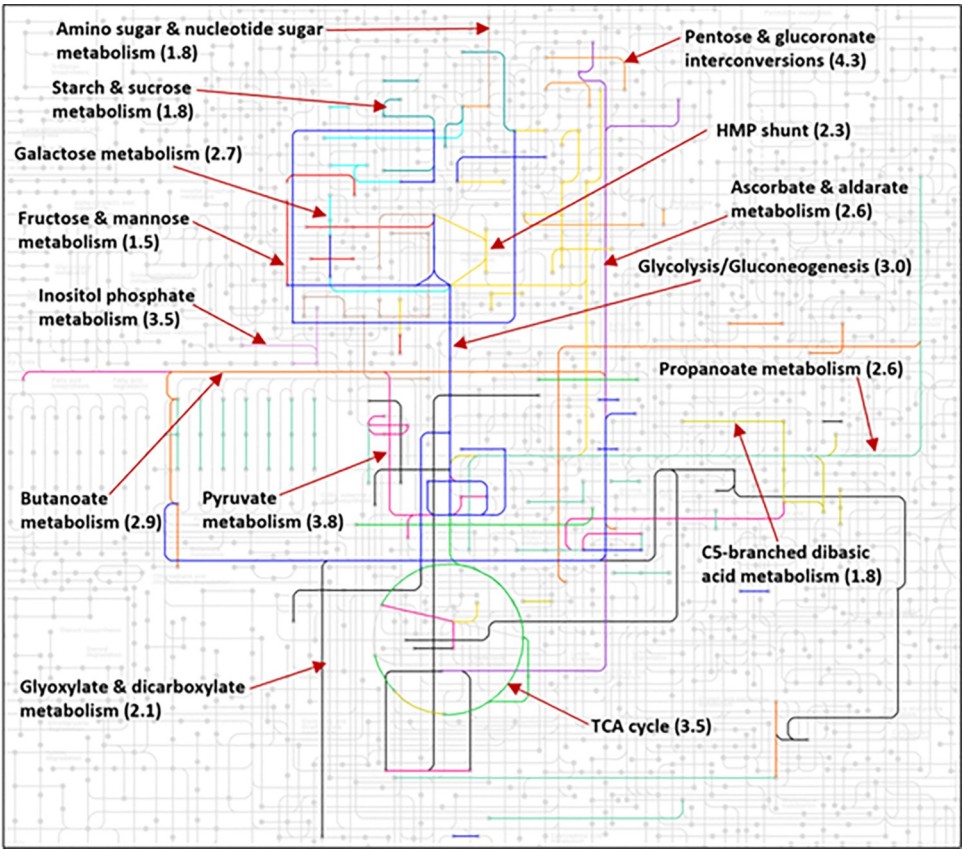

**Fig 7. *E. coli* carbohydrate pathway is the largest of all pathways.** With 11 different sub-systems, we found that *E. coli*'s carbohydrate metabolic pathway has the largest number of enzymes (146), proteins (299) and protein-enzyme interactions (378) compared to other pathways such as lipid (33,108,119) or amino acid metabolism (111,200,261) [3, 61]. The number in parentheses are average interactor counts per enzyme (*density values*) for that pathway. The *E. coli* metabolome map can be found at https://www.genome.jp/pathway/map01100.

secondary (secondary interactor—primary interactor/enzyme) and primary interactions (primary interactor—metabolic enzyme) (**Fig 12**). However, given the uncertainty of these calculations based on the predicted structures and interfaces, we decided not to use these results in our final analysis.

## 3 Results & discussion

### 3.1 Selecting 16 PEIs as probable regulators of metabolism

We illustrate in **Fig 4** how we started from 378 PEIs and reached the number of 16 PEIs which were later reported as potential metabolism regulators. The figure shows how we filtered out PEIs during each analysis.

### 3.2 The carbohydrate metabolism protein interactome

There are 242 enzymes which metabolize carbohydrates (**Table 1**), including 146 enzymes (60%) which bind 299 other proteins (or interactors). Multiple enzymes can have a common interactor or multiple interactors binding one enzyme (**Table 2**). A summary of these is provided in **Table 2**.

Carbohydrate metabolism enzymes interact primarily with metabolic enzymes. This is not surprising as proteins of similar functions often bind to each other [41, 50]. It is well-known

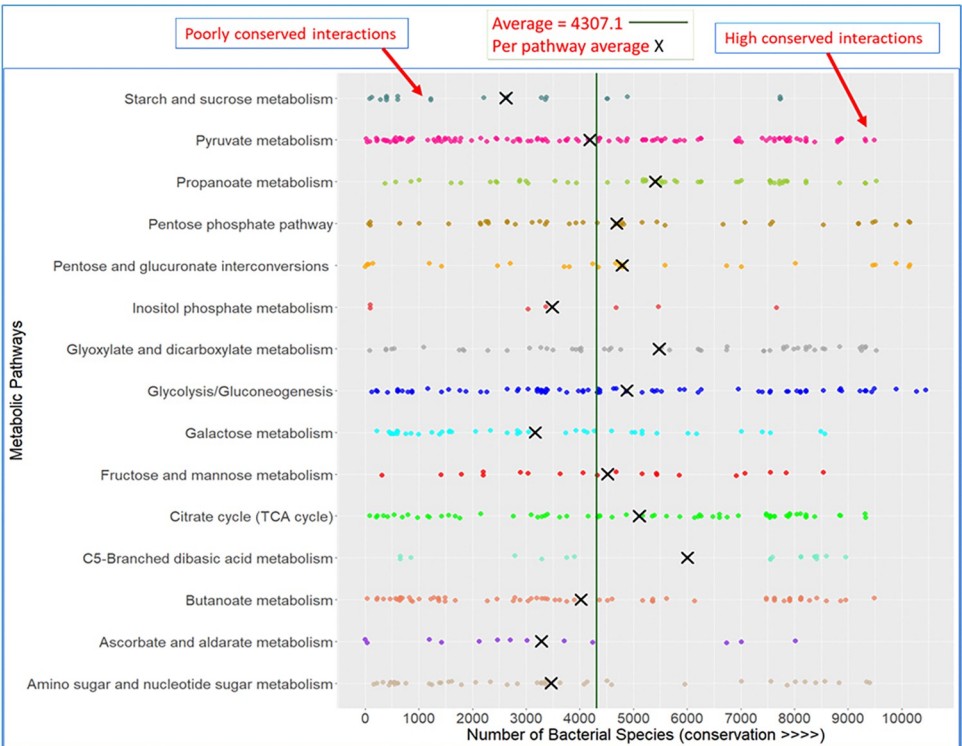

**Fig 8. Conservation of interactions in CHM.** *E. coli* pathways including central carbohydrate pathways have more conserved PEIs than others. The average conservation value per pathway is shown by a X and for all pathways, it is shown by a vertical line. Each dot represents the number of bacterial genomes where a PEI co-occurs as orthologs, so, in total, there are 378 dots for 378 PEIs.

that carbohydrate metabolism is controlled through protein-protein interactions between the enzymes of this pathway. For instance, interactions between GltA of the TCA cycle and PykF of glycolysis [51], Gnd of the Pentose-phosphate shunt and PckA of the TCA cycle are well known [52]. Carbohydrate enzymes also bind 12 enzymes not involved in the 11 *E. coli* metabolic pathways [2], such as Chaperonin GroEL [53], Arginine import ATP-binding proteins (AraG) [54], Flagellum-specific ATP synthase (FliI) [55] and Multiphosphoryl transfer protein 1 (FryA) [56].

45 interactors of carbohydrate enzymes are metabolic proteins but not enzymes (indicated by not having an EC number). These proteins mainly include transcription factors, chaperones and ribosomal proteins. There are 59 non-metabolic proteins interacting with metabolic enzymes and thus may play a role in metabolism (**Fig 5**) [57].

Surprisingly, *E. coli* carbohydrate enzymes also interact with 54 proteins that are either uncharacterized or that lack annotation (S1 Table in S1 File). Among them, 3 unknown proteins show potentially relevant expression levels: YadG, YeeD and DcrB. These three proteins are at least 3 times more abundant than their enzyme partners GadB, MurA and GloC respectively. Investigating these interactions involving unknown proteins experimentally may shed light on their metabolic or non-metabolic roles [20, 58, 59].

## 3.3 Count of protein-protein interactions per carbohydrate pathway

The protein-enzyme interactions were classified into different metabolic pathways based on the pathway of the enzyme. For example, PEI GlpK-Eno will be part of glycolysis, given that

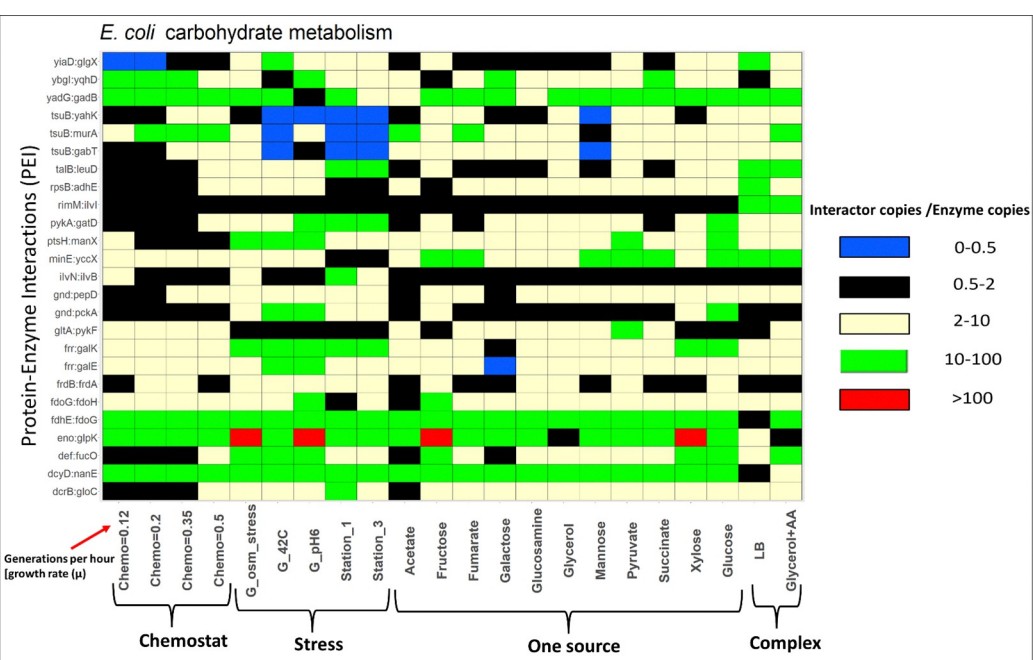

**Fig 9. Impact of growth condition on abundance and ratios of PEIs.** The ratio (abundance of interactor/abundance of enzyme) for 25 *E. coli* PEIs (**y axis**) is shown for 22 different *E. coli* environments (**x axis**). Ratios were divided into 5 different ranges shown as a colored brick. Blue bricks denote a ratio of 0–0.5 while a green brick shows values from 10–100. A blue or black brick indicates that the enzyme is more or equally abundant than the interactor in that media. A green, yellow or red brick shows that the interactor is more abundant than the enzyme in that media. G_ means glucose media, so, for example G_42C means glucose media at 42˚C.

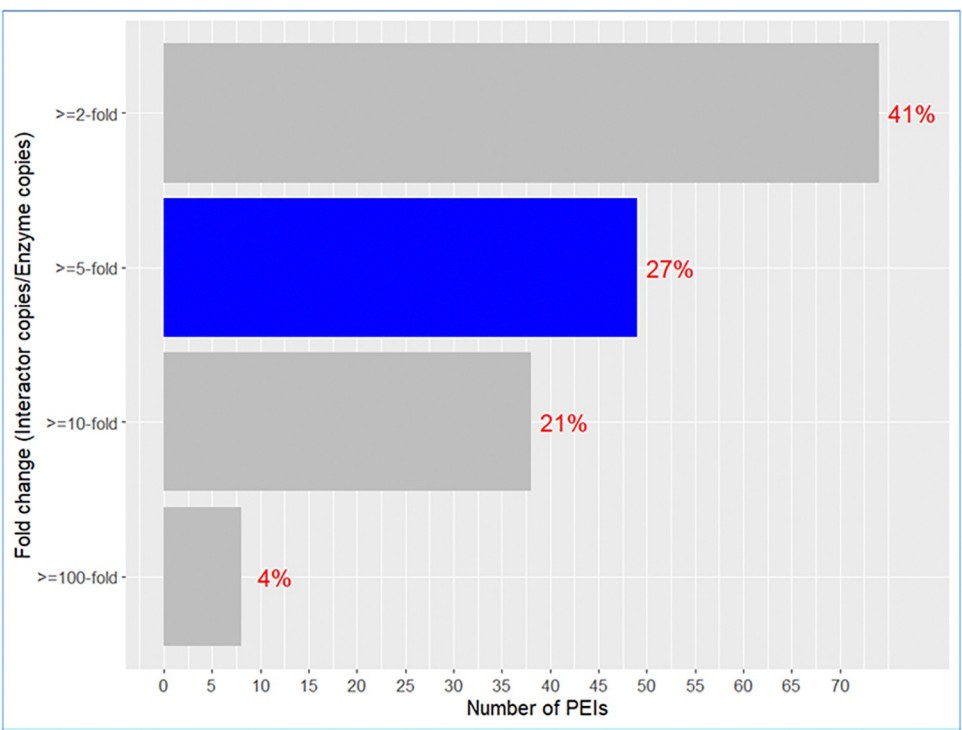

**Fig 10. Number of CHM PEIs at specific interactor/enzyme ratios.** 27% of *E. coli* PEIs (48 PEIs out of 179 PEIs) showed fold change of > = 5 in at least one cell growth condition (blue bar) [29]. These cases were investigated in more detail to delineate their abundance properties in 22 different metabolic states (**Fig 8**) [29].

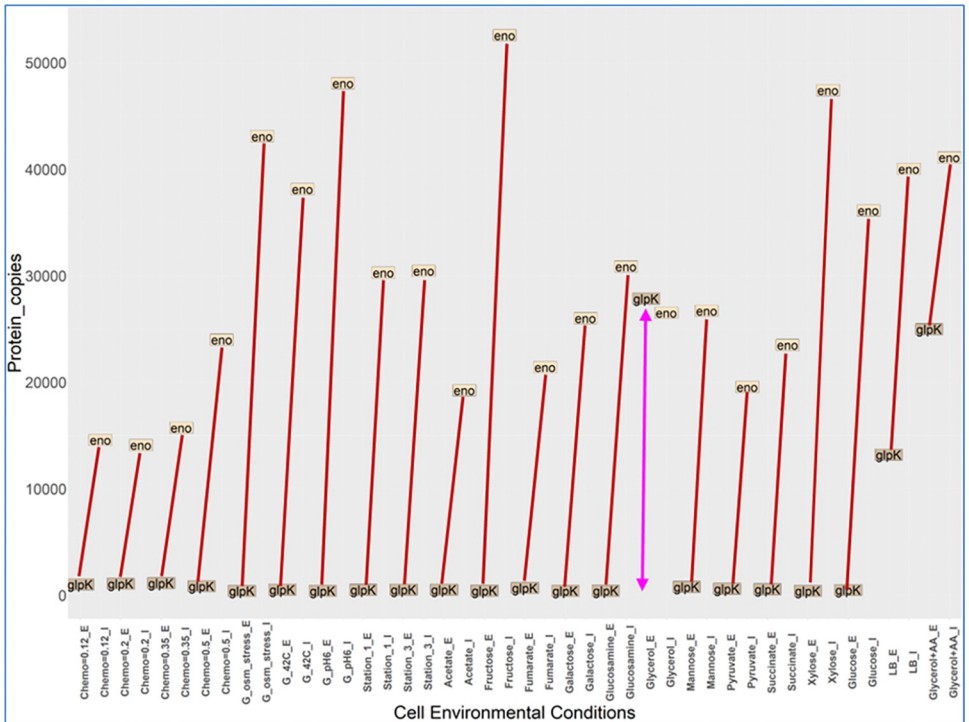

**Fig 11. Abundance of enolase (Eno) and its binding enzyme GlpK in different metabolic states.** GlpK; glycerol kinase is a glycerol processing enzyme which increases its abundance in glycerol and glycerol + AA media. Its binding partner Eno may not be responsible for this increase in abundance of GlpK as it does not show itself any condition specific abundances. On the x axis, G means glucose media and G_pH6 means glucose media at pH6. For each of the 22 conditions, there are duplicates of that such as Fumarate_E and Fumarate_I. _E means the abundance of Enzyme in that condition whereas _I mean abundance of Interactor in that condition.

Enolase is a glycolytic enzyme. We found that pathways with more metabolic enzymes bind more proteins and thus, have more PEIs (**Fig 6**). For instance, glycolysis and TCA cycle have 26 and 19 enzymes and 83 and 70 PEIs, respectively. Metabolites of pathways such as glycolysis and TCA often function as branchpoints to other metabolic pathways, hence they affect these other pathways too, such as amino acid or lipid metabolism. Therefore, these pathways have large numbers of enzymes and it is very likely that PPIs regulate these pathways [60].

It is not surprising that pathways with more enzymes bind more interactors. Therefore, we used another measure to reflect on the count of PEIs per metabolic pathway which is how many interactors bind a metabolic enzyme on average. The *density value* is equal to the number of interactors divided by the number of enzymes and we calculated it for each of the 11 carbohydrate pathways. The average of those values was 2.68. Glycolysis, TCA cycle and pyruvate metabolism pathways had density values > 3. Inositol phosphate metabolism, galactose metabolism and the pentose & glucoronate interconversions pathway had density values of 3.5, 2.9 and 4.3 respectively (**Fig 7**). This means that these pathways with a low number of enzymes (2, 14 and 6) bind a relatively high number of interactors (7, 38 and 26 respectively). Overall, out of 11 *E. coli* metabolic pathways [30] including lipid, amino acid or nucleotide metabolism pathways, we find that carbohydrate metabolism has the largest number of enzymes, reactions, interactors and thus, PEIs. **Fig 7** illustrates how much space carbohydrate metabolism pathway takes in the entire *E. coli* metabolome with *density values* shown in () after the pathway names.

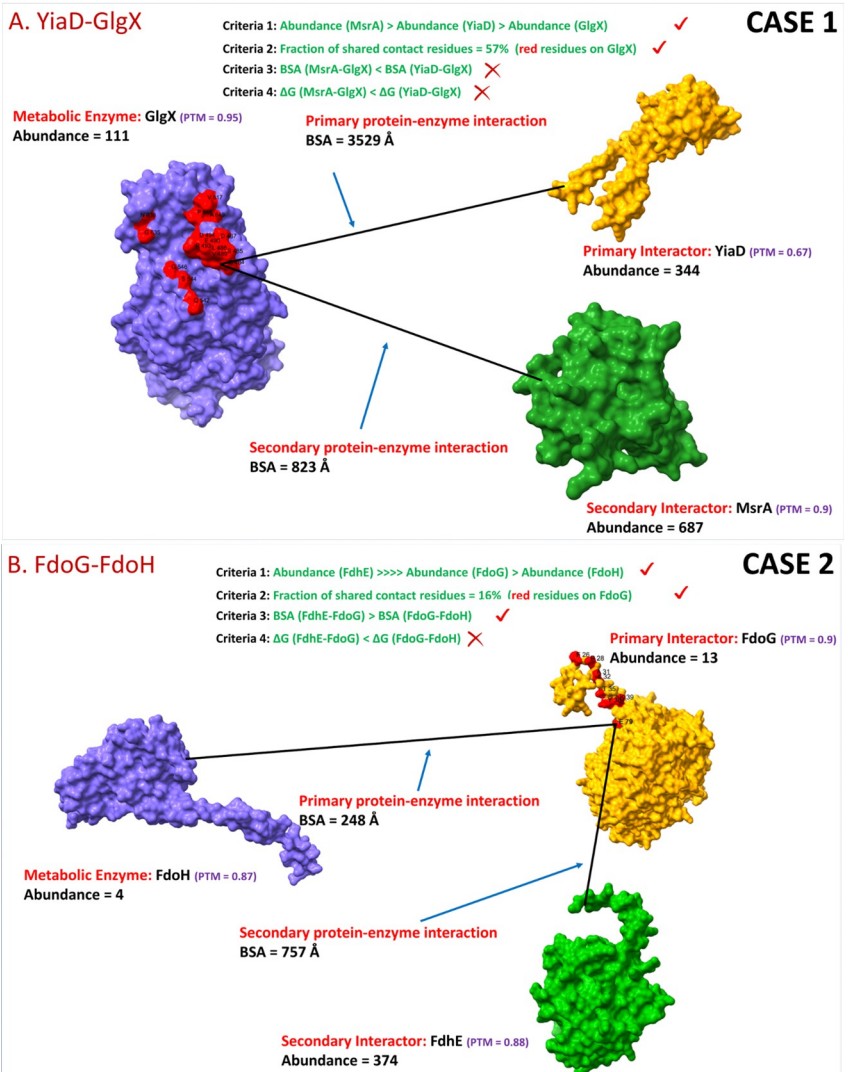

**Fig 12.** Primary interactions (A) YiaD-GlgX (Case 1) and (B) FdoG-FdoH (Case 2) may get affected by their secondary interactors. The structures were obtained from ColabFold [42] and the PTM scores [43] were above 0.6 for all of them reflecting high confidence in predicting the structures of these proteins accurately. The residues highlighted in red on primary interactor are binding sites where all secondary interactors and metabolic enzyme bind that protein.

## 3.4 Conservation of carbohydrate PEIs

A protein with orthologs in a large number of species is more conserved and therefore, may play important biological functions [62]. Thus, we tested the conservation of *E. coli* PEIs by counting in how many bacterial genomes these PEIs co-occur as orthologs. Overall, we considered 10000 bacterial genomes listed in the EggNOG database [40]. For each 378 PEIs, we calculated the number of bacterial genomes where both the interactor and its binding metabolic enzyme co-occur. The average number of bacterial genomes where *E. coli* PEIs are found is 4307 (green line in **Fig 8**). Therefore, in 43% of bacterial genomes, proteins are predicted to bind carbohydrate enzymes.

If we focus on individual pathways, the average conservation value per pathway (**X** in **Fig 8**) is higher than 4307 for the central carbohydrate pathways of glycolysis, TCA cycle and HMP

shunt. If metabolic regulation happens at protein-protein interaction (PPI) levels, we assume it will happen more for these pathways [3, 63]. Pathways which *E. coli* uses only in specific metabolic environments like starch and sucrose, ascorbate and aldarate, inositol phosphate metabolism and amino sugar and nucleotide sugar metabolism were less conserved as the average conservation value (**X**) was less than 4307 for these pathways (**Fig 8**). Overall, all pathways span a large range of conservation, i.e. some may be highly conserved in one species but not in others. This is not surprising, given that closely related species will have higher conservation levels than distantly related, ecologically divergent species.

Of the 378 carbohydrate metabolism PEIs in *E. coli* (**Table 2**), on average, there are 150 PEIs found in a bacterial genome. We list the top 5 bacterial genomes in each of these three groups in **Table 3**. For example, 354 *E. coli* PEIs are found as orthologs in *Shigella sonnei* 53G which is the causative agent of bacterial dysentery [64]. *Shigella* and other γ-proteobacteria are close relatives of *E. coli*, therefore, the number of co-occurring PEIs is high in them. With increasing phylogenetic distance, the number of orthologous pairs decreases, but equally important, the similarity of proteins also decreases, so many of the predicted interactions based on sequence similarity may be lost in evolution. Nevertheless, we expect that PEIs can regulate metabolism not only in *E. coli* but in many other species where they co-occur. Hence, experimental investigation of *E. coli* PEIs may help us to 1) develop novel therapeutic strategies [65], 2) understand novel metabolic functions of human microbiome [66] and 3) design new applications in biotechnology [67].

## 3.5 Abundances of metabolic enzymes and their binding partners suggest novel metabolic regulation

We wondered how many of the 378 PEIs may be metabolic regulators in *E. coli*. Importantly, regulation often depends on the stoichiometry of an enzyme and its regulator [68]. Hence, we obtained abundance values in the form of protein copies per cell for the *E. coli* proteins from Schmidt et al [29] who determined abundance of approx. 2600 *E. coli* proteins in 22 different environments. These conditions included multiple carbon sources, complex media, chemostat cultures as well as cultures under stress (**Fig 9 x axis**). Schmidt et al [29] provides abundance data for 179 PEIs out of 378 that we identified earlier. For an interactor to regulate the functioning of the enzyme when it binds it, it can be assumed that it should be in higher amounts to do it when it binds the metabolic enzyme [69]. Following this assumption, we calculated the ratio of Interactor/Enzyme copies for each of the 179 PEIs. Then, we computed how many PEIs had ratios of at least 100, 10, 5 and 2 in at least one environmental condition (**Fig 10**). We focused on an arbitrary ratio of Interactor/Enzyme copies $> = 5$ in at least one growth media (**Fig 10**). Much higher thresholds (e.g., 100-fold) result in only 7 PEIs for the next sets of investigations and a very low threshold (2-fold) may lead to false positives (73) [70]. Therefore, we somewhat arbitrarily chose 5-fold as threshold which gave us 48 PEIs (out of 179) for further investigations.

In 48 PEIs the number of interactor molecules was at least 5 times than the number of enzyme molecules in at least one growth media. To identify which of the 48 PEIs have conditional effects (possible regulatory) we identified those PEIs for which the ratio of interactor copies/enzyme copies is $> = 5$ in one growth media and $< = 1$ in a different growth media, indicating a condition-dependent change in the PEI. Condition-dependent changes to PEIs are a significant indicator associated with regulation of metabolism [28, 71]. Out of 48 PEIs, 25 PEIs (**Fig 9**) showed condition-dependent changes in abundance ratios. A large range of ratios, from $> 100$ copies of interactor:enzyme (shown in red in **Fig 9**) to $< 0.5$ (indicating more enzyme than interactor, shown in blue in **Fig 9**) was observed.

### 3.6 Some enzyme abundances change independent from interactors

Absolute values of interactor and enzyme abundance were also analyzed in the 22 different growth conditions. We found 4 PEIs (among 25) where only the enzyme changed its abundance. These 4 PEIs were 1) Frr-GalE, 2) Frr-GalK, 3) YadG-GadB and 4) Eno-GlpK.

Frr is a ribosomal protein which binds the enzymes GalE and GalK. GalK is galactokinase (GalK) and UDP-glucose 4-epimerase (GalE), and these enzymes get activated in media with galactose [72]. We observed that across all 22 growth conditions, Frr is consistently higher in abundance than GalK and GalE except galactose media where both these enzymes become more abundant than Frr (**S1** and **S2** **Figs**).

GadB is usually expressed at low levels while its binding partner; YadG is highly abundant across all growth media. Glucose media at pH6 ("G_pH6) is an exception with GadB still being lower in abundance than YadG but the difference in abundance between these two proteins becomes very small due to upregulation of GadB (**S3 Fig**). GadB expression is known to be upregulated when *E. coli* faces acidic stress [73].

The Eno-GlpK interaction is an enzyme-enzyme interaction between glycolytic enzyme Enolase and Glycol kinase GlpK. In 21 of the 22 growth conditions, GlpK is lower in abundance than its interactor, Eno, with growth on glycerol being the one exception [74] (**Fig 11**). In complex media Glycerol + AA, GlpK shows an increase in abundance. In these 4 cases the metabolic enzyme increased abundance levels when the metabolic function was required independently from interactor abundances.

### 3.7 Enzymes and interactors compete with other proteins to bind to each other

Enzymes are not just regulated by single interactors but yet other proteins may compete or interfere with these interactions. We called these proteins "secondary interactors" and focused on 21 such PEIs [75]. These 21 interactions, involved primary interactors with at least 5-fold excess over the enzyme in at least one growth media but switched to a different ratios in other growth media (**Section 3.5**). Note that 48 PEIs had primary interactors with at least 5-fold excess over their enzyme in at least one growth media but only 25 PEIs among them reversed their stoichiometry in some condition(s), so we focused on those remaining 23 PEIs (48–25). There are 2 cases to consider (**Fig 12**):

**Case 1.** In those 23 PEIs, there can be a secondary interactor with abundance higher than both the metabolic enzyme and the primary interactor, so, they may occupy most of the binding sites on the enzyme and hence, the primary interactor may not be able to bind the metabolic enzyme (**Fig 12A**).

**Case 2.** The primary interactor is more abundant than the metabolic enzyme, but the abundance of the secondary interactor is much higher than the primary interactor, so it will occupy most of the binding sites on the primary interactor and the metabolic enzyme may not get chance to bind the primary interactor (Fig 12B).

In order to investigate the cases described above, we obtained the abundance values of secondary interactors of enzymes (Case 1) and primary interactors (Case 2) in Glucose and LB media and investigated the impact of abundance, binding sites, surface area, and binding energy:

**Abundance:** In order to determine which PEIs get affected by secondary interactions, we used the abundance of secondary interactors of an enzyme. For instance, Case 1 is represented by MsrA (687 copies per cell) which binds GlgX (111 copies) (**Fig 12A and Table 4**). LeuB (4588), IlvH (1533) and MqsA (34) bind IlvI (441) as its secondary interactors (**Table 4**). For Case 2, we calculated the abundance ratios of (secondary interactor / primary interactor) /

(primary interactor / metabolic enzyme). If this ratio is $>= 5$, we considered that secondary interactors might hinder the primary interactor–enzyme binding (**Section 3.5**). Examples of this cases are PEIs FdoG-FdoH (**Fig 12B**) or Def-FucO (see **Table 4** for other examples).

**Binding sites:** Secondary interactors can block primary interactors if they compete for the same binding (**Fig 12**). Therefore, for the 5 PEIs in **Table 4**, we attempted to determine the binding sites among enzymes and (primary and secondary) interactor [76] as described in (**Section 2.7**) [42, 77]. The main results of this analysis are summarized in **Table 5**.

Overall, the PEIs YiaD-GlgX (**Fig 12B**), RimM-IlvI and DcrB-GloC tested positive for **Criteria 1** (Abundance) and **Criteria 2** (Binding sites). However, they did not give the results we hypothesized in **Criteria 3** (BSA) and **Criteria 4** (ΔG), hence we excluded this data from further analysis. We must keep in mind that we included the effects of all secondary interactors (cumulative abundance) in **Criteria 1** but in **Criteria 2**, **3** and **4**, we focused on only the most highly abundant secondary interactor binding either the enzyme (Case 1) or primary interactor (Case 2) (**Table 5**). Therefore, a single secondary interaction may not show a high BSA but if cumulative effects of all secondary interactors is considered, these 3 PEIs may test positive in Criteria 2 or 3 but it is very difficult to determine and would make things bit complex. We predict PEIs Def-FucO and FdoG-FdoH (**Fig 12B**) have more chances of getting obstructed by secondary interactors RplQ/D and FdhE based on these criteria. It is very hard to predict the actuality of secondary interactions computationally, however, in this study for the first time, we report it as a novel metabolic regulatory mechanism affecting the interaction between a primary interactor and metabolic enzyme.

Note that it is possible that secondary interactors may be both competitors [75] or activators [78]. For example, secondary interactor FdhE may also act as a promoter of Formate dehydrogenase complex formation [79].

## 3.8 Predicting potential regulators of *Escherichia coli* carbohydrate metabolism

**3.8.1 Multi-level regulation of metabolism.** The fumarate reductase complex is a good example for multi-level regulation of protein complexes in metabolism. FrdB interacts with metabolic enzyme FrdA to form the fumarate reductase enzyme complex [84]. FrdA is the catalytic subunit and FrdB is the regulatory subunit [86]. The original studies did not investigate the endogenous stoichiometry in any details, but Schmidt et al. [29] found both of them at high abundance in fumarate media (**Fig 11A**). Transcription factor ArcA (Aerobic respiration control protein) causes high production of these two proteins when a major nutrient is fumarate (**Fig 11B**) [85].

**3.8.2 Enzyme-enzyme interactions.** Enzymes catalyzing one metabolic reaction often bind and regulate other enzymes (**Fig 13**). Out of 16 PEI candidates (**Table 6**), 4 of them are enzyme-enzyme interactions. 1) GltA (TCA cycle) binds PykF (glycolysis) (**Fig 13**) [51, 87]. 2) Gnd pentose-phosphate shunt) binds PckA (TCA cycle) [52, 89]. 3) Gnd also binds enzyme PepD (amino acid metabolism). Both these proteins are highly expressed in *Salmonella Typhi*, the causative agent of typhoid in humans [90]. 4) Glycolytic enzyme PykA interacts with GatD (galactose pathway) and both are inhibited by DicF which is an sRNA helping *E. coli* adapt to changing environments [88]. Overall, we found that metabolic enzymes prefer other metabolic enzymes as their binding partners (**Fig 5**) [50, 98–100].

**3.8.3 Protein-protein interactions in sulfur metabolism.** Among the 16 PEI regulators, one protein interacts with three different carbohydrate enzymes. The putative sulfur carrier protein TsuB binds MurA (peptidoglycan biosynthesis), GabT (Butanoate metabolism) and YahK (glycolysis). TsuB has been reported to physically interact with many proteins to

**Table 6. *E. coli* PEIs predicted as potential regulators of metabolism.**

| Protein-Enzyme Interaction (PEI) | Carbohydrate metabolism pathway | Reference |
|---|---|---|
| **Multi-level Regulation of Metabolism** | | |
| IlvN-IlvB | C5-branched dibasic acid metabolism/ Butanotate metabolism | [80, 81] |
| FdhE-FdoG | Glyoxylate & dicarboxylate metabolism | [82, 83] |
| FrdB-FrdA | Citrate cycle/ Butanoate metabolism/ Pyruvate metabolism | [84–86] |
| **Enzyme-Enzyme interactions** | | |
| GltA-PykF | Glycolysis/ Pyruvate metabolism | [51, 87] |
| PykA-GatD | Galactose metabolism | [88] |
| Gnd-PckA | Glycolysis/ Citrate cycle/ Pyruvate metabolism | [52, 89] |
| PepD-Gnd | Pentose phosphate pathway | [90] |
| **Sulfur metabolism** | | |
| TsuB-MurA | Amino sugar & nucleotide sugar pathway | [91–94] |
| TsuB-YahK | Glycolysis | |
| TsuB-GabT | Butanoate metabolism | |
| **PEI-specific literature evidence** | | |
| TalB-LeuD | C5-branched dibasic acid metabolism | [95] |
| PtsH-ManX | Fructose & mannose metabolism/ Amino sugar & nucleotide sugar metabolism | [5, 10, 96] |
| YbgI-YqhD | Propanoate metabolism | [97] |
| **Novel Regulators** (No literature evidence) | | |
| RpsB-AdhE | Pyruvate metabolism/ Glycolysis/ Butanoate metabolism | N/A |
| MinE-YccX | Pyruvate metabolism | N/A |
| DcyD-NanE | Amino sugar & nucleotide sugar pathway | N/A |

facilitate thiosulfate uptake in *E. coli* [91, 92] which is an important process during carbohydrate metabolism regulation [93, 94]. Therefore, further experimental investigations on these 3 PEIs involving TsuB may give us more insights on what role sulfur assimilation plays during carbohydrate metabolism in *E. coli*.

**3.8.4 Protein-enzyme interaction specific literature evidence.** The functions of 3 PEIs are supported by published data: TalB-LeuD, YbgI-YqhD and PtsH-ManX. PtsH (= Hpr)

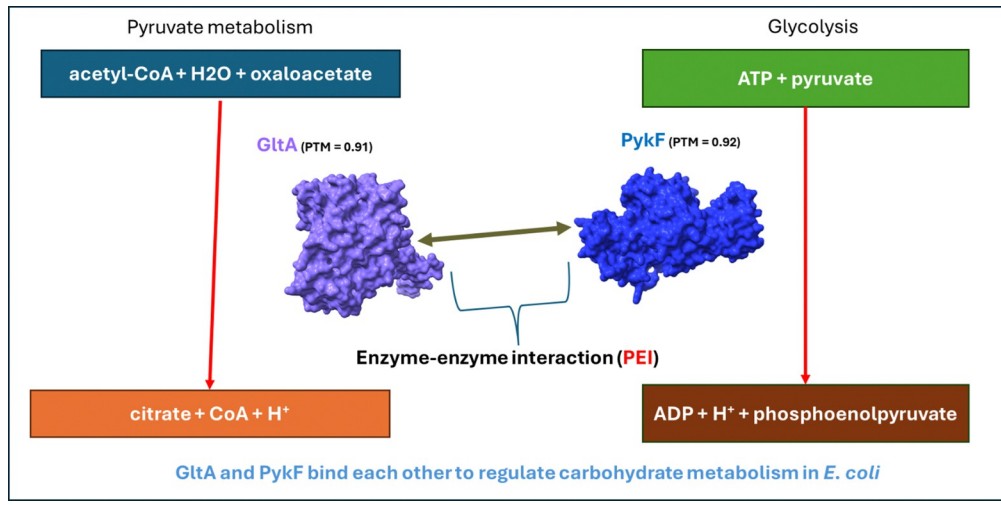

**Fig 13. Enzyme-enzyme interaction between citrate synthase (GltA) and pyruvate kinase (PykF).**

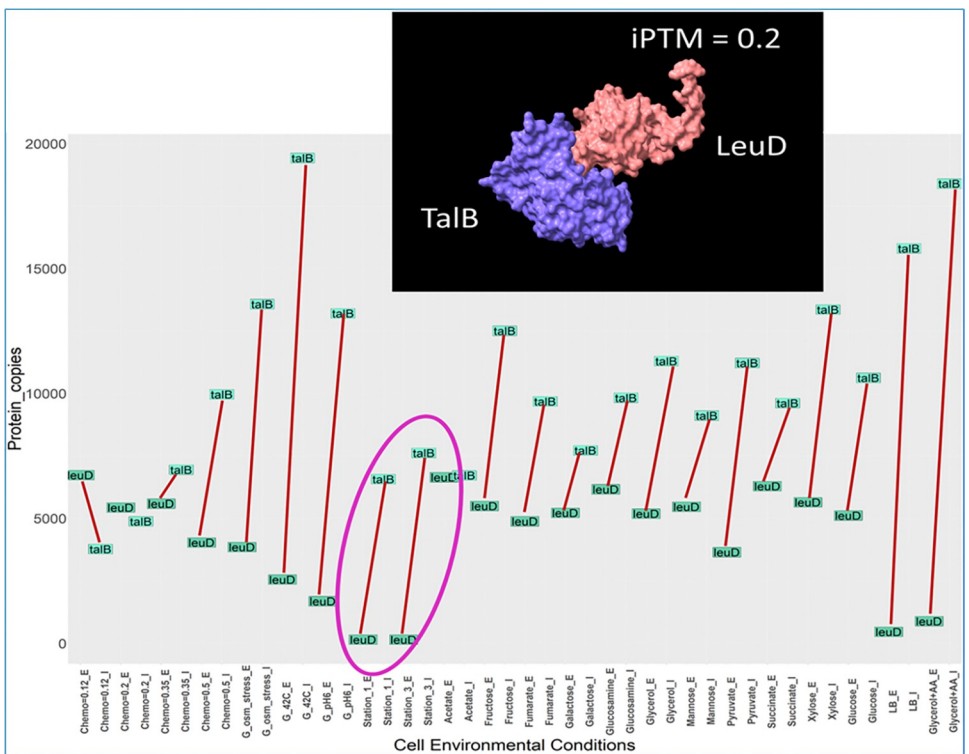

**Fig 14. Abundance of binding proteins TalB and LeuD in 22 different growth conditions.** Both these interacting proteins become low in abundance due to growth inhibitor ClpX degrading them [95] when *E. coli* goes through stationary phase highlighted in pink circle. G_ means glucose media, e.g., G_42C is glucose media at 42˚C. E indicates abundance of enzyme and I = abundance of interactor in that media.

binds another phosphotransferase system protein, ManX and both these proteins are inhibited by the transcription repressor Mlc. Mlc is known to repress the expression of Pts proteins when *E. coli* does not need glycolysis in low glucose conditions [5, 96]. ManX and PtsH both are Pts proteins [101]. PtsH protein interacts with and allosterically regulates multiple *E. coli* metabolic enzymes [10].

Next, interactor YbgI binds the metabolic enzyme YqhD and both YbgI and YqhD are known to increase their amounts in LB condition when *E. coli* cells experience high vanillin in their cell environments [97].

**Fig 14** illustrates both Interactor TalB and its binding partner, enzyme LeuD decrease their abundance when *E. coli* goes through stationary phases. This is happening due to the function of a third protein, ClpX [95]. This protein degrades both TalB and LeuD when *E. coli* switches to stationary phase implying these proteins facilitate cell growth.

## 4 Conclusion

This project is the first attempt to look at regulation of carbohydrate metabolism via the binding partners of metabolic enzymes in *Escherichia coli*. Although highly desirable, the experimental investigations of the predicted PEIs was beyond the scope of this study. However, we selected 16 PEIs as good candidates for experimental analysis with a good chance of these predictions being metabolism regulators. Our study predicts 3 PEIs as metabolism regulators previously not studied: RpsB-AdhE, DcyD-NanE & MinE-YccX (**Table 6**) and many others involving uncharacterized proteins (**Fig 5**). Therefore, this opens doors for experimental

biologists to investigate these PEIs further and to develop novel applications of these PEIs in *E. coli* and other bacteria. In this study, we showed how metabolism regulation can be dependent on quantities of binding proteins helping *E. coli* survive in multiple cell environments. Apart from that, based on features including abundance, their predicted binding affinity with other proteins etc., we reported potentially new protein-interaction regulatory mechanisms such as secondary interactions that can either obstruct a primary protein-enzyme interaction or increase its strength by forming metabolic enzyme complexes. Lastly, we were able to uncover new protein-enzyme interactions that are likely to function during sulfur metabolism, gene regulation-metabolism crosstalk and stationary phase in *E. coli*.

## Supporting information

**S1 Fig. Absolute abundance of PEI Frr-GalK in 22 different growth conditions.**
(TIF)

**S2 Fig. Absolute abundance of PEI Frr-GalK in 22 different growth conditions.**
(TIF)

**S3 Fig. Absolute abundance of PEI YadG-GadB in 22 different growth conditions.**
(TIF)

**S4 Fig. Enzyme IlvB depends (enzyme of butanoate and C5-branched dibasic acid metabolism) on its interactor IlvN for its function.**
(TIF)

**S5 Fig. Enzyme FdoG, (dehydrogenizing formate) and its interactor, FdhE simultaneously increase abundance dramatically in LB but are less abundant in other growth conditions.**
(TIF)

**S1 File.**
(XLSX)

## Author Contributions

**Conceptualization:** Stephen S. Fong, Peter Uetz.

**Formal analysis:** Shomeek Chowdhury.

**Investigation:** Shomeek Chowdhury, Peter Uetz.

**Methodology:** Shomeek Chowdhury, Peter Uetz.

**Project administration:** Stephen S. Fong, Peter Uetz.

**Supervision:** Stephen S. Fong, Peter Uetz.

**Visualization:** Peter Uetz.

**Writing – original draft:** Shomeek Chowdhury.

**Writing – review & editing:** Stephen S. Fong, Peter Uetz.

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
