## [Decision Letter · Decision Letter 0]

23 Jun 2024

PONE-D-24-16447The protein interactome of Escherichia coli carbohydrate metabolism.PLOS ONE

Dear Dr. Uetz,

Thank you for submitting your manuscript to PLOS ONE. After careful consideration, we feel that it has merit but does not fully meet PLOS ONE’s publication criteria as it currently stands. Therefore, we invite you to submit a revised version of the manuscript that addresses the points raised during the review process.

 Manuscript has been reviewed by 2 subject experts. You may go through their comments. Both have shown interest in the work and appreciated in the importance of hypothesis. They have given their comments and suggestions. I could gather that the conclusion drawn from provided data is not convincing and therefore, it needs a complete rewriting. Reviewer 1 did not find much of new information and the importance of study on regulation of enzyme activity by protein -protein interaction has come nicely. This point should be brought out in revision.

We look forward to receiving your revised manuscript.

Kind regards,

Hari S. Misra, Ph.D.

Academic Editor

PLOS ONE

Additional Editor Comments (if provided):

Reviewers' comments:

Reviewer's Responses to Questions

**Comments to the Author**

1. Is the manuscript technically sound, and do the data support the conclusions?

Reviewer #1: Partly

Reviewer #2: Yes

2. Has the statistical analysis been performed appropriately and rigorously? 

Reviewer #1: N/A

Reviewer #2: N/A

3. Have the authors made all data underlying the findings in their manuscript fully available?

Reviewer #1: Yes

Reviewer #2: Yes

4. Is the manuscript presented in an intelligible fashion and written in standard English?

Reviewer #1: No

Reviewer #2: Yes

5. Review Comments to the Author

Reviewer #1: Summary:

This study aimed to investigate how carbohydrate metabolism (CHM) in E. coli is regulated by their interaction properties with other proteins and their quantities. By collecting published protein abundance data from reference [15] and PPI data from the IntAct database from references [31-33], they built a list of 378 PEIs (protein enzyme interactions) for carbohydrate metabolism enzymes. Then, each component of PEIs was assigned with their normalized abundance values (protein copies/cell) under 22 different growth conditions. By interpretation only, they showed examples of PEI changes in response to growth conditions, possibly due to tug-of-war bindings through secondary interactors or the direct completion for the shared binding sites. Finally, the consequent pair-wise PEI stoichiometry (the abundance ratio between enzymes and their interactors) was used to “predict/hypothesize” the potential regulators of E. coli carbohydrate metabolism.

Generally, this is a pure in silico study using published data. The purpose of this study is straightforward. The PEIs summarized are intuitive. However, I feel this is an incomplete and immature research and the manuscript is not well prepared. Moreover, I do not think this study provides sufficient new knowledge. Finally, I cannot agree that this study directly addresses the primary question of “how carbohydrate metabolism (CHM) in E. coli is regulated by their interaction properties with other proteins and their quantities” which they claimed in the Abstract and Conclusion.

Major comments:

1. “PEIs” is a relatively new conceptual PPI category. It focuses on enzymes and their interactors. Although not strongly mentioned in the manuscript, it seems there is a hypothesis that the interactors are likely to be the regulators of the enzyme, and quantitatively or qualitatively changing the interactors might be a potential mechanism to regulate the enzyme. However, I feel disappointed to find out that this study only generates a list of PEIs, which makes no clear difference from its sources IntAct database. Moreover, in the whole manuscript, I could feel the authors tried very hard to search for many works of literature to support or imply the potential of their findings related to PEIs. However, there is still nothing new from the PEIs in this study even if some literature had previously identified the same enzyme-protein interaction as their PEIs list. In the end, it turned out that the interpretation of the results/discussion looked like just a literature review.

2. From the M/M to Results section, it looks like a direct copy and paste from a student thesis. The Results are fragmented and lacking clear logical connections. At least, each experiment or analysis should be composed of (1) scientific motivations/questions, (2) methodologies to investigate the issues (3) results, and (4) interpretations. Moreover, the figure legends are dispersed throughout the manuscript and it is difficult to find them. It is disrespectful to reviewers who have to wade through it.

To me, the Discussion is relatively better in the scientific context. I would suggest to rewrite a mixed Results/Discussion section.

3. It is confusing how to define and classify the “primary” and “secondary” interactors of the same enzyme/protein. Is it based on ratio or absolute abundance?

4. P11, “Enzymes and interactors compete with other proteins to bind to each other”. Honestly, I am not convinced by simple discussions to claim that interactors compete with each other when only “predicted” overlapped binding sites are shown.

5. Abstract, P12, Conclusions. “E. coli protein-enzyme interactions may regulate carbohydrate metabolism”, I do not see any solid and direct evidence to support this conclusion solely according to the data generated in this study.

The only way I can think of to improve this study is to perform some wet experiments to verify some previously unpublished and uninvestigated PEIs and then to prove that the interactor plays roles in the regulation of the paired enzyme.

Minor comments:

1. Table 1: change “KEGG_Num” to “KEGG Identifier”.

2. Intuitively, “non-metabolic enzyme” sounds make no sense to me. I assume all enzymes work for metabolisms of the organism. Please clarify this definition, and provide some examples showing an enzyme with non-metabolic functions.

3. Please provide more detailed information in M/M for the “Conservation of PEIs” analysis. It is also good to know the classification of bacterial OG species used.

4. It would be also good to add one more column in Table S1 after “Enzyme” and “Interactor” to specify their functions, e.g., dehydrogenase, converting A from B, a regulator of enzyme X, etc.

5. P5, “Number of PEIs scale with pathway size.” It is not surprising at all. PPI is universal. A bigger pathway composed of more enzymes of course gets more interactors. I will not feel this is an informative analysis unless “PEI density (numbers of interactors per enzyme)” is used for comparison and then some specific pathways are found to have unexpected higher or lower PEI density.

6. P12, L396, the M/M for using ChimeraX and comparing overlapped binding sites is missing.

7. Are the two interacting proteins' protein structures based on AlphaFold completely? In other words, are their interaction based on a predicted model or real protein crystal structures?

8. P9, except for the galactose examples, how do you explain the other “inverse stoichiometry” cases?

9. Fig S1-S5, what do “_E” ,“_I”, and “G_” stand for in the growth conditions X-axis?

10. Fig S1-S5 needs to be re-prepared. Some gene name tags close to the Y-axis and right side border are truncated.

11. Fig 10, move the statement “Schmidt et al [15] PEIs count = 179 in the figure to the figure legend or direct remove it.

Reviewer #2: The authors studied over 300 protein-enzyme interactions (PEIs) for carbohydrate metabolism (CHM) enzymes in E. coli and found extensively interactions with other metabolic enzymes as well as non-metabolic proteins. The authors showed that many PEIs vary with metabolic conditions, showing significant regulatory dynamics. The authors also investigated secondary interactors that could compete with PEIs and offered some likely mechanisms of regulatory functions of the enzyme interaction partners.

Major concerns:

1. The authors should justify the use of the IntAct database to retrieve PEIs as there are many PPI databases.

2. The authors studies possible effects of secondary interactors under the assumption of competing roles. Secondary interactors could also facilitate the interaction of a PEI if they form stable multi-subunit complexes. This possibility should be mentioned.

3. Details of how the contacts are calculated in Table 4 and Figure 6 should be given in materials and methods. Are they based on experimental structures or compute-modeled structures? If experimental structures were used, the PDB codes of the structures shown in Figure 6 should be given.

4. While the secondary interaction and a PEI may share a certain fraction of contacting residues, they may not interfere with each other. Superposition of secondary interaction complex and the PEI complex to detect spatial clashes would be more suitable to infer competing roles of a secondary interactor.

5. The pLDDT score is residue based. Are the pLDDT scores in Figures 7 and 8 the average of all pLDDTs? pLDDT cannot be used to quantify the confidence of the complex. The authors can use AlphaFold3 to model the complex and use the PTM and iPTM scores as confidence measures of the overall complex and the interface respectively.

Minor points:

Page 2, line 11, Escherichia coli: change to italic. Line 22, Hpr (italic): change to regular font HPr.

Page 3, line 56. It is not clear what “non-spoke expanded interactions” means.

Page 8, line 257 mentioned “PEIs in red in Table 5”. However, there is no red marking in Table 5.

To make the supplementary tables more informative, add gene names in Tables S1, S2 and S3 and add UniProt accessions in Table S4.

6. PLOS authors have the option to publish the peer review history of their article (what does this mean?). If published, this will include your full peer review and any attached files.

Reviewer #1: No

Reviewer #2: No

---

## [Author Response · Author response to Decision Letter 0]

13 Oct 2024

Please see the rebuttal letter for a detailed description of changes and improvements in this revised manuscript.

---

## [Editor Report · Decision Letter 1]

3 Nov 2024

PONE-D-24-16447R1The protein interactome of Escherichia coli carbohydrate metabolism.PLOS ONE

Dear Dr. Uetz,

Thank you for submitting your manuscript to PLOS ONE. After careful consideration, we feel that it has merit but does not fully meet PLOS ONE’s publication criteria as it currently stands. Therefore, we invite you to submit a revised version of the manuscript that addresses the points raised during the review process and by editor as appended below.

Figures and legends in revised manuscript seems to have got inadvertently duplicated. Authors did not check the final version before submission. They are suggested to revise the revised manuscript, scrutinize carefully and then submit as a revision.

We look forward to receiving your revised manuscript.

Kind regards,

Hari S. Misra, Ph.D.

Academic Editor

PLOS ONE
---

## [Author Response · Author response to Decision Letter 1]

12 Nov 2024

Reviewer comments has been addressed in the Response to Reviewers word file. The line numbers mentioned in Response to Reviewers document are in accordance with Manuscript w figures and highlights (reviewer comments) word file. We have added Manuscript w figures and highlights (reviewer comments) so that reviewers can see what revisions we have made to the manuscript by referring to Response to Reviewers file and Manuscript w figures and highlights (reviewer comments) simultaneously.

The Manuscript with Data Av statement (w/o figures) file is the manuscript file with no highlights and figures. We have uploaded 19 figures in total (14 (Figs 1-14) + 5 (Figs S1-S5)). There is also a word file with Suppl Figures (S1 - S5): Supplementary Figures 1-5

In the latest decision, editor mentioned that we have some figures duplicated. Actually, the figures from the older version (original version) were also added by mistake. We have removed them this time and the manuscript should consist of only figures of the revised version.

Nov 12 update: Editor mentioned to upload a manuscript file with no highlights and track changes. It is called Manuscript w figures.docx. We have uploaded it.

---

## [Editor Report · Decision Letter 2]

22 Nov 2024

The protein interactome of Escherichia coli carbohydrate metabolism.

PONE-D-24-16447R2

Dear Dr. Uetz,

We’re pleased to inform you that your manuscript has been judged scientifically suitable for publication and will be formally accepted for publication once it meets all outstanding technical requirements.

Kind regards,

Hari S. Misra, Ph.D.

Academic Editor

PLOS ONE
---

## [Editor Report · Acceptance letter]

3 Dec 2024

PONE-D-24-16447R2 

PLOS ONE

Dear Dr. Uetz, 

I'm pleased to inform you that your manuscript has been deemed suitable for publication in PLOS ONE. Congratulations! Your manuscript is now being handed over to our production team.

Kind regards, 

on behalf of

Professor Hari S. Misra 

Academic Editor

PLOS ONE